# Clinical utility of inflammatory biomarkers in COVID-19 in direct comparison to other respiratory infections—A prospective cohort study

Maurin Lampart[1], Núria Zellweger[2], Stefano Bassetti[3], Sarah Tschudin-Sutter[4,5], Katharina M. Rentsch[6], Martin Siegemund[2,4], Roland Bingisser[7], Stefan Osswald[1], Gabriela M. Kuster[1], Raphael Twerenbold[1,8,9]*

1 Department of Cardiology and Cardiovascular Research Institute Basel (CRIB), University Hospital Basel, University of Basel, Basel, Switzerland, 2 Intensive Care Unit, University Hospital Basel, Basel, Switzerland, 3 Division of Internal Medicine, University Hospital Basel, Basel, Switzerland, 4 Department of Clinical Research, University of Basel, Basel, Switzerland, 5 Division of Infectious Diseases and Hospital Epidemiology, University Hospital Basel, Basel, Switzerland, 6 Laboratory Medicine, University Hospital Basel, Basel, Switzerland, 7 Emergency Department, University Hospital Basel, Basel, Switzerland, 8 University Center of Cardiovascular Science and Department of Cardiology, University Heart and Vascular Center Hamburg, University Medical Center Hamburg-Eppendorf, Hamburg, Germany, 9 German Center for Cardiovascular Research (DZHK), Partner Site Hamburg-Kiel-Lübeck, Hamburg, Germany

* r.twerenbold@uke.de

**Data Availability Statement:** In order to allow, facilitate, and accelerate urgently needed clinical studies during the early phase of the COVID-19

## Abstract

### Background

Inflammatory biomarkers are associated with severity of coronavirus disease 2019 (COVID-19). However, direct comparisons of their utility in COVID-19 versus other respiratory infections are largely missing.

### Objective

We aimed to investigate the prognostic utility of various inflammatory biomarkers in COVID-19 compared to patients with other respiratory infections.

### Materials and methods

Patients presenting to the emergency department with symptoms suggestive of COVID-19 were prospectively enrolled. Levels of Interleukin-6 (IL-6), c-reactive protein (CRP), procalcitonin, ferritin, and leukocytes were compared between COVID-19, other viral respiratory infections, and bacterial pneumonia. Primary outcome was the need for hospitalisation, secondary outcome was the composite of intensive care unit (ICU) admission or death at 30 days.

### Results

Among 514 patients with confirmed respiratory infections, 191 (37%) were diagnosed with COVID-19, 227 (44%) with another viral respiratory infection (viral controls), and 96 (19%)

pandemic, the responsible ethics committee (Ethics Committee Nordwest- und Zentralschweiz (EKNZ), Hebelstrasse 53, 4056 Basel, Tel. 061 268 13 50, Fax 061 268 13 51, Email: eknz@bs.ch; EKNZ identifier 2020-00566) waived the need for a study-specific informed consent. Instead, they gave permission to include patients in this study based on the signature of an unspecific general consent, which allows to analyze clinical parameters and blood remains that are collected during clinical routine and to contact patients up to 30 days after hospital discharge. However, this general consent does not permit to make patient-level data publicly available, not even in an anonymized fashion, as these are highly sensitive and potentially identifying patient data from a single-center study obtained during a short period of time. However, in case of a request for a scientific collaboration, data sharing could be allowed under the umbrella of a site-by-site data transfer agreement granting adequate data protection and confidentiality. Access to the data is restricted by the review board of the COVIVA Study. Data requests may be directed to Professor Raphael Twerenbold (raphael.twerenbold@usb.ch), or to Gian Völlmin (gian.voellmin@usb.ch), a representative of the data access committee.

**Funding:** This study was supported by research grants from the Schweizerische Herzstiftung (Swiss heart foundation) (https://www.swissheart.ch, RT), the Cardiovascular Research Institute of Basel (CRIB) (http://www.crib-usb.ch, RT), and Roche Diagnostics (https://diagnostics.roche.com, RT). The funders had no role in study design, data collection and analysis, decision to publish, or preparation of the manuscript.

**Competing interests:** RT has received speaker honoraria/consulting honoraria from Abbott, Amgen, Astra Zeneca, Roche, Siemens, Singulex, and Thermo Scientific BRAHMS, outside the submitted work. GK has received consultant fees from Janssen, outside the submitted work. Authors not named here have disclosed no competing interests. This does not alter our adherence to PLOS ONE policies on sharing data and materials.

with bacterial pneumonia (bacterial controls). All inflammatory biomarkers differed significantly between diagnoses and were numerically higher in hospitalized patients, regardless of diagnoses. Discriminative accuracy for hospitalisation was highest for IL-6 and CRP in all three diagnoses (in COVID-19, area under the curve (AUC) for IL-6 0.899 [95%CI 0.850–0.948]; AUC for CRP 0.922 [95%CI 0.879–0.964]). Similarly, IL-6 and CRP ranged among the strongest predictors for ICU admission or death at 30 days in COVID-19 (AUC for IL-6 0.794 [95%CI 0.694–0.894]; AUC for CRP 0.807 [95%CI 0.721–0.893]) and both controls. Predictive values of inflammatory biomarkers were generally higher in COVID-19 than in controls.

## Conclusion

In patients with COVID-19 and other respiratory infections, inflammatory biomarkers harbour strong prognostic information, particularly IL-6 and CRP. Their routine use may support early management decisions.

## 1. Introduction

The current global pandemic with severe acute respiratory syndrome coronavirus 2 (SARS-CoV-2) represents a massive burden on healthcare systems worldwide. Many clinical predictors for poor outcome in patients with Coronavirus disease 2019 (COVID-19) have been discovered, including high levels of inflammatory biomarkers such as interleukin-6 (IL-6), c-reactive protein (CRP), ferritin, and procalcitonin (PCT) [1–14]. However, most of these studies had a retrospective design and lacked an adequate control group to directly compare findings observed in COVID-19 to patients with acute respiratory infections other than COVID-19.

IL-6 is secreted by macrophages as a pro-inflammatory cytokine. It is an important mediator of the acute phase reaction [15–17] and plays a major role in the development of cytokine release syndrome (CRS), when dysregulated [18–20]. CRS is characterised by a clinical phenotype of systemic inflammation, multi-organ failure and death, caused by an extreme increase in the inflammatory response of multiple cytokines. Triggers of CRS are heterogenic and may be of rheumatologic, oncologic, or infectious origin [18, 19]. CRS is an important cause of poor outcome in COVID-19 [2, 21–23]. Less is known about the role of IL-6 in respiratory infections of other cause. However, previous studies show an association of IL-6 with more severe disease [20, 24–26]. Normal values of IL-6 are below 10 pg/ml.

CRP is an acute-phase inflammatory protein produced in the liver [27, 28]. Its production is induced primarily but not exclusively by IL-6 [28–30]. CRP is an opsonin and therefore binds to the surface of cells. This activates the complement system which leads to phagocytosis by macrophages [28, 31]. CRP is a well-established and broadly used predictor of poor outcome for infections of any origin and therefore used in COVID-19, community acquired pneumonia and viral respiratory infections [10, 32–37]. Normal values for CRP are below 5 mg/l.

PCT is a precursor peptide of calcitonin. It is produced by the parafollicular cells of the thyroid and the neuroendocrine cells of the lung tissue. It rises significantly in infections of bacterial origin, and is therefore used as a guide for antibiotic therapy in patients with infections of unknown origin [38]. High PCT levels due to infections, however, are not followed by an

increase in calcitonin. Increased levels of PCT in COVID-19 patients have been described and an association with disease severity has also been shown [3, 9, 39, 40]. Normal values for PCT are below 0.1 ng/ml. Bacterial infection is very likely if PCT values exceed 1 ng/ml.

Ferritin is one of the most important storage proteins for iron, but also an acute-phase inflammatory protein that is elevated under various conditions including inflammation, coronary artery disease, and malignancy [41, 42]. Already prior to SARS-CoV-2 pandemics, ferritin has been identified as a predictor of poor-outcome in acute respiratory distress syndrome [43]. Similarly, it is also increased in COVID-19 and correlates with disease severity [9, 44, 45]. Normal values for ferritin are approximately between 20–300 μg/l and differ between men and women. Low levels of ferritin are an indicator of iron deficiency, high levels of ferritin can be a sign of hemochromatosis, malignancy, or infections.

Leukocyte levels often increase during infections due to the release of several molecules, as growth or survival factors, adhesion molecules and various cytokines released during activation of immune system. Most bacterial infections are associated with neutrophilic leukocytosis. Neutrophilia occurs from both upregulated bone marrow production and the release of neutrophils from the endothelium. Generally, most viruses lead to relative lymphocytosis, while only a few viruses causing lymphopenia, such as SARS-CoV-2 [10, 46, 47].The causes for lymphopenia in COVID-19 have not yet been conclusively determined. Possible mechanisms include, but are not limited to, SARS-CoV-2-induced apoptosis of lymphocytes via the angiotensin converting enzyme 2, CRS-induced apoptosis of lymphocytes, and antibody-dependent killing of SARS-CoV-2-infected lymphocytes [48]. The leukopenia is mostly driven by lymphopenia, as SARS-CoV-2 binds to the angiotensin converting enzyme 2 (ACE2), which is located on most lymphocytes [49, 50]. Leukocytosis is defined by an increase in the WBC count of more than 11,000 cells/microL.

We aim to investigate the diagnostic accuracy and prognostic utility of the above-described inflammatory biomarkers (IL-6, CRP, PCT, ferritin, and leukocytes) to predict hospitalisation and outcome in cases with COVID-19 and compare them with cases with respiratory infections other than COVID-19.

## 2. Materials and methods

### 2.1. Study design, population, and inclusion criteria

The COronaVIrus surviVAl (COVIVA, ClinicalTrials.gov NCT04366765) is a prospective, observational cohort study including consecutive patients aged minimally 18 years presenting with clinically suspected or confirmed SARS-CoV-2 infection to the emergency department (ED) of the University Hospital Basel, Switzerland, during the first wave of COVID-19 pandemic between 23 March 2020 and 7 June 2020. All patients underwent nasopharyngeal SARS-CoV-2 polymerase chain reaction (PCR) swab tests. Patients were considered SARS-CoV-2 positive if one or multiple SARS-CoV-2 PCR swab tests performed at day of ED presentation or within two weeks prior to or post ED presentation were positive in combination with clinical signs and symptoms. The remainders with only negative SARS-CoV-2 swab test results were considered as controls. All participating patients or their legally authorized representatives consented by signing a local general consent form. This study was conducted according to the principles of the Declaration of Helsinki and approved by the local ethics committee (EKNZ identifier 2020–00566). The authors designed the studies, gathered, and analysed the data according to the STROBE guidelines, vouched for the data and analysis, wrote the paper, and decided to submit it for publication (**S1 Table**).

## 2.2. Clinical assessment

All subjects underwent a thorough clinical assessment by the treating physician according to local standard operating procedures. Vital signs such as heart rate, blood pressure, oxygen saturation and respiratory rate were documented in every patient.

## 2.3. Blood sampling

Blood samples were drawn in both cases and controls at time of ED presentation. CRP, ferritin, and leukocytes without further white blood cell differential were measured in fresh samples as part of clinical routine of the recruiting hospital using Roche analyzers (Roche Diagnostics, Rotkreuz, Switzerland). For research purposes, serum samples were collected and stored at -80°C. IL-6 and PCT was measured in frozen serum samples in a dedicated external laboratory (Roche Diagnostics, Penzberg, Germany). Treating physicians were blinded for IL-6 and PCT, but not the remaining investigational inflammatory biomarkers.

## 2.4. Follow-up

Patients were followed-up by telephone or in written form by research physicians or study nurses thirty days after discharge. Information about current health, hospitalisations and adverse events was collected using a predefined questionnaire. Records of hospitals and primary care physicians, as well as national death registries, were screened for additional information, if applicable.

## 2.5. Outcomes

The primary outcome was defined as the need for hospitalisation at time of ED presentation. The secondary outcome was defined as the composite of intensive care unit (ICU) admission or all-cause death at 30 days. For additional analyses, disease severity was categorized into four groups (outpatients, normal ward survivors, ICU survivors, and decedents at 30 days).

## 2.6. Adjudication of final diagnosis

The adjudication of the final diagnosis that led to the index ED presentation and the clinical suspicion of COVID-19 was performed in each patient by a pool of five trained physicians, who reviewed all medical data available (e.g., chest x-ray, routine laboratory parameters) including 30-day post-discharge follow-up information and chose from a predefined list of diagnoses what best fit each patient. Each adjudication was primarily assigned by one physician per patient, only. However, all uncertain cases were discussed collectively within the adjudicating team and final decision was made in the consensus by majority vote. Predefined main categories included but were not limited to COVID-19, non-SARS-CoV-2 infections (e.g., other respiratory, gastrointestinal, urogenital), cardiovascular disease (acute coronary syndrome, rhythm disorder, congestive heart failure, pulmonary embolism), other pulmonary non-infectious disease (e.g., lung tumor, asthma, chronic obstructive pulmonary disease) and neurologic disease (e.g., stroke, seizure). For this analysis, we only used respiratory infections other than COVID-19 as controls. All cases with viral respiratory infections other than COVID-19 served as viral controls, cases with pneumonia served as bacterial controls. The distinction between bacterial pneumonia and viral respiratory infection was primarily based on clinical examination (e.g., rales, fever, tachypnoea) and particularly radiological findings (e.g., lobar or interstitial pneumonic infiltrates in the x-ray or CT scan of the lungs). No specific pathogen distinction to identify the underlying bacterium or virus (e.g., bacterium isolation on sputum sample, urinary antigen positivity for pneumococcus or legionella, virus isolation on

multiplex PCR) was systematically performed as part of clinical routine and was therefore largely missing.

## 2.7. Statistical analysis

In the present analysis, cases with COVID-19 were compared with the two control groups: First, with viral controls (cases with respiratory infections other than COVID-19) and second, with bacterial controls (cases with bacterial pneumonia). Data are expressed as medians and interquartile range (IQR) for continuous variables, and as numbers and percentages (%) for categorical variables. Missing values were not imputed. All variables were compared by Mann-Whitney-U test for continuous variables with binary outcomes, Kruskal-Wallis test for continuous variables with multiple outcomes, and Pearson's $\chi^2$ or Fisher's exact test for categorical variables, as appropriate. Levels of inflammatory biomarkers were displayed using boxplots and compared between groups according to the primary and secondary outcomes as well as disease severity. We assessed the discriminative performance by the receiver operating characteristic curve (ROC) and the area under the curve (AUC) for the primary and secondary outcomes. A value of 0.5 indicates no predictive ability, a value of 0.8 is considered good, and 1.0 is perfect. We performed logistic regression analysis for the primary outcome and Cox proportional regression analysis for the secondary outcome in a univariable and a multivariable approach. For multivariable analysis we performed a stepwise backwards selection. To achieve a normal distribution, we used log transformation on all inflammatory biomarkers for all regression analysis. For the secondary outcome in COVID-19, we performed event curve analysis, using a Kaplan Meier estimator, using the median for the respective biomarker in COVID-19 as cut-off value. For comparison of event rates, we used the log-rank test and the hazard ratio (HR). P-values smaller than 0.05 were considered significant. No correction for multiple testing was applied. Statistical analysis was performed using R software package, version 4.0.5, and IBM SPSS Statistics for Windows, version 27.0 (IBM Corp., Armonk, NY, USA).

## 3. Results

### 3.1. Baseline characteristics

Overall, 1202 cases presenting with symptoms suggesting COVID-19 were screened and 1086 were enrolled in this study from 23 March 2020 to 7 June 2020. Follow-up at 30 days after discharge was completed in 1081 cases. COVID-19 was confirmed in 191 (37%) cases and 323 cases were diagnosed with an acute respiratory infection of other cause, of which 227 (44%) were of viral (viral controls), and 96 (19%) of bacterial origin (bacterial controls, **Fig 1**). The baseline demographic and clinical characteristics of COVID-19 cases and both controls are shown in **Table 1**. Bacterial controls were significantly older than COVID-19 patients (72 years [IQR 58–80] vs. 57 years [IQR 44–69], p<0.001), whereas viral controls were significantly younger (52 years [IQR 35–64] vs. 57 years [IQR 44–69], p = 0.004). Therefore, bacterial controls had in general equal or more comorbidities than COVID-19 patients. Numbers of missing values for the displayed variables are depicted in **S2 Table**.

### 3.2. Inflammatory biomarkers in COVID-19 cases and controls

As displayed in **Fig 2**, median levels of inflammatory biomarkers differed significantly between COVID-19 patients and both control groups. In bacterial controls, IL-6, CRP, and PCT were higher (IL-6 80.90 pg/ml [IQR 34.38–278.59], CRP 73.3 mg/l [IQR 14.6–134.1], PCT 0.149 ng/ml [IQR 0.052–0.481]), than in COVID-19 (IL-6 20.77 pg/ml [IQR 4.56–46.48], CRP 28.9 mg/l

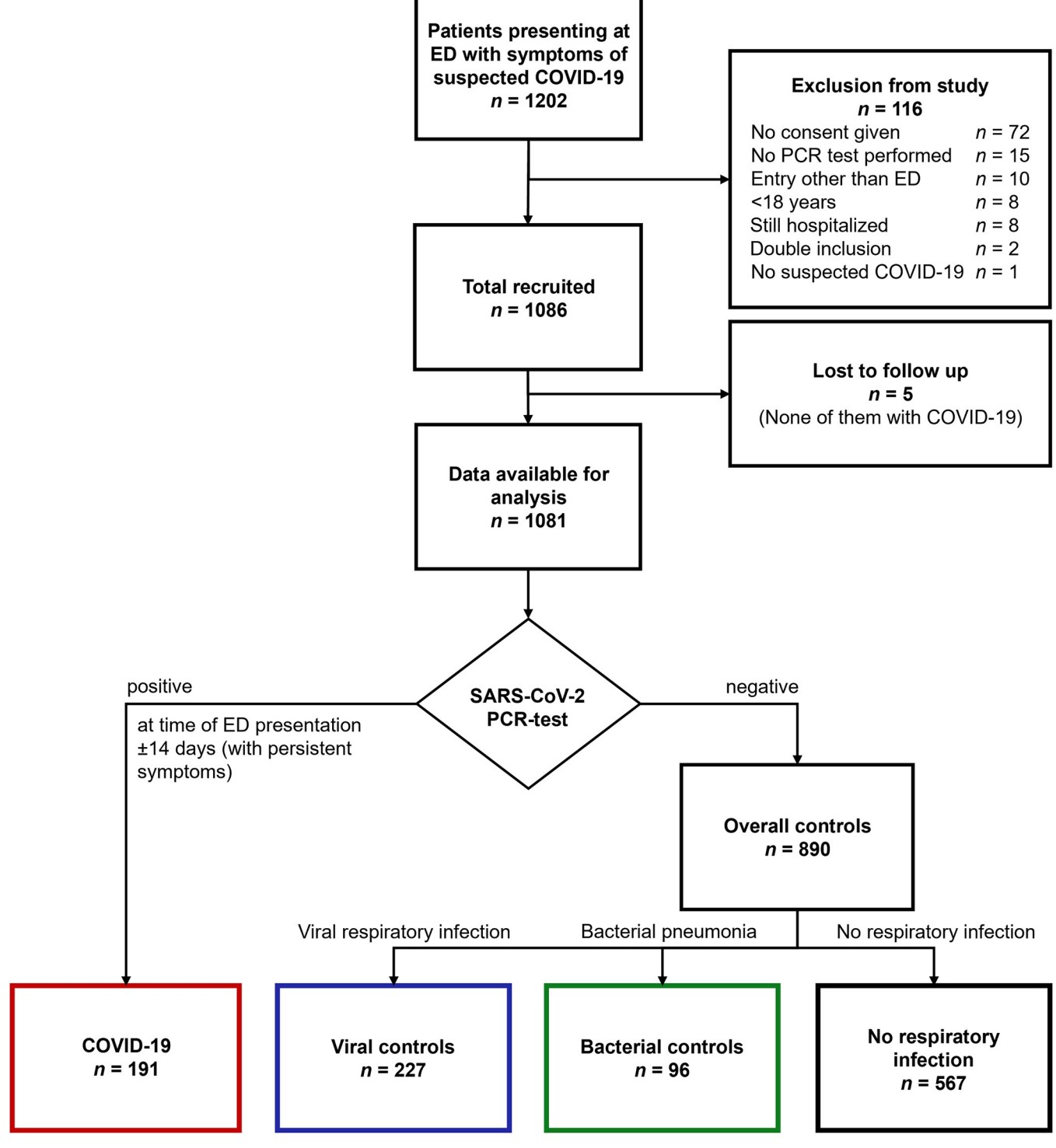

**Fig 1. Flow chart.** ED = emergency department, COVID-19 = coronavirus disease 2019, PCR = polymerase chain reaction.

[IQR 2.6–73.4], PCT 0.046 ng/ml [IQR 0.024–0.116]) with p-values <0.001 for all comparisons. However, they were lower in viral controls (IL-6 4.61 pg/ml [IQR 2.09–16.51], CRP mg/l 3.3 [IQR 0.9–15.0], PCT 0.030 ng/ml [IQR 0.014–0.056]), than in COVID-19 with p-values

**Table 1. Baseline characteristics in COVID-19 and controls.**

| Measures | COVID-19 | Viral controls | p-value[a] | Bacterial controls | p-value[b] |
|---|---|---|---|---|---|
| | n = 191 | n = 227 | | n = 96 | |
| Demographics | | | | | |
| Age—years | 57 [44–69] | 52 [35–64] | 0.004 | 72 [58–80] | <0.001 |
| Female | 84 (44) | 105 (46) | 0.641 | 37 (39) | 0.379 |
| Comorbidities—no (%) | | | | | |
| Cardiac disease[c] | 38 (20) | 51 (22) | 0.522 | 41 (43) | <0.001 |
| • Valvular cardiopathy | 8 (4) | 9 (4) | 0.908 | 7 (7) | 0.265 |
| • Coronary artery disease | 21 (11) | 26 (11) | 0.882 | 17 (18) | 0.113 |
| • Prior myocardial infarction | 9 (5) | 12 (5) | 0.789 | 10 (10) | 0.067 |
| • Atrial fibrillation | 9 (5) | 10 (4) | 0.881 | 23 (24) | <0.001 |
| Hypertension | 81 (42) | 87 (38) | 0.396 | 55 (57) | 0.170 |
| Overweight | 74 (39) | 69 (30) | 0.073 | 22 (23) | 0.007 |
| Diabetes | 36 (19) | 27 (12) | 0.088 | 24 (25) | 0.152 |
| Ever smoker | 58 (30) | 105 (46) | 0.001 | 54 (56) | <0.001 |
| Pneumopathy[d] | 37 (19) | 88 (39) | <0.001 | 39 (41) | <0.001 |
| • Asthma | 25 (13) | 41 (18) | 0.165 | 13 (14) | 0.915 |
| • COPD | 9 (5) | 37 (16) | <0.001 | 21 (22) | <0.001 |
| Hepatopathy | 14 (7) | 23 (10) | 0.315 | 14 (15) | 0.051 |
| CKD | 26 (14) | 14 (6) | 0.010 | 25 (26) | 0.009 |
| Stroke | 10 (5) | 9 (4) | 0.534 | 10 (10) | 0.104 |
| Cancer | 17 (9) | 12 (5) | 0.147 | 18 (19) | 0.016 |
| Immunodeficiency | 11 (6) | 11 (5) | 0.677 | 14 (15) | 0.012 |
| Symptoms at ED—(%) | | | | | |
| Symptom duration before ED—days | 7 [3–11] | 5 [2–10] | 0.067 | 3 [2–7] | <0.001 |
| Cough | 126 (66) | 182 (80) | 0.001 | 60 (63) | 0.562 |
| Dyspnea | 81 (42) | 136 (60) | <0.001 | 49 (51) | 0.166 |
| Vital signs at ED | | | | | |
| Systolic BP—mmHg | 135 [122–148.5] | 142 [126–156] | 0.004 | 132 [120–152] | 0.667 |
| Diastolic BP—mmHg | 82 [71–90] | 82 [74–89] | 0.412 | 80 [70–86] | 0.112 |
| Heart rate—/min | 89 [80–103] | 88 [76–101] | 0.298 | 95 [80–110] | 0.053 |
| Blood oxygen saturation—% | 97 [94–98] | 97 [96–98] | 0.001 | 95 [92–97] | <0.001 |
| Respiratory rate—/min | 20 [16–24] | 18 [15–21] | 0.001 | 22 [19–27] | <0.001 |
| Temperature - ˚C | 37.1 [36.8–38.0] | 36.9 [36.5–37.3] | <0.001 | 37.4 [36.9–38.3] | 0.027 |
| Laboratory parameters at ED | | | | | |
| IL-6—pg/ml | 20.77 [4.56–46.48] | 4.61 [2.09–16.51] | <0.001 | 80.90 [34.38–278.59] | <0.001 |
| CRP—mg/l | 28.9 [2.6–73.4] | 3.3 [0.9–15.0] | <0.001 | 73.3 [14.6–134.1] | <0.001 |
| PCT—ng/ml | 0.046 [0.024–0.116] | 0.030 [0.014–0.056] | <0.001 | 0.149 [0.052–0.481] | <0.001 |
| Ferritin - µg/l | 387 [164–823] | 137 [76–238] | <0.001 | 266 [153–435] | 0.008 |
| Leukocytes—G/l | 6.27 [4.95–8.34] | 8.34 [6.85–10.70] | <0.001 | 10.89 [8.65–14.55] | <0.001 |

[a] p-value for comparison of COVID-19 with viral controls

[b] p-value for comparison of COVID-19 with bacterial controls

[c] cardiac disease includes valvular cardiopathy, coronary artery disease, prior myocardial infarction, and atrial fibrillation

[d] pneumopathy includes asthma and COPD.

Continuous variables were compared using the Mann-Whitney-U test, and categorical variables using the Pearson χ2 test or Fisher's exact test, as appropriate. Values are numbers (percentages) or median [interquartile range]; COVID-19 = coronavirus disease 2019, COPD = chronic obstructive pulmonary disease, CKD = chronic kidney disease, ED = emergency department, BP = blood pressure, IL-6 = interleukin-6, CRP = c-reactive protein, PCT = procalcitonin.

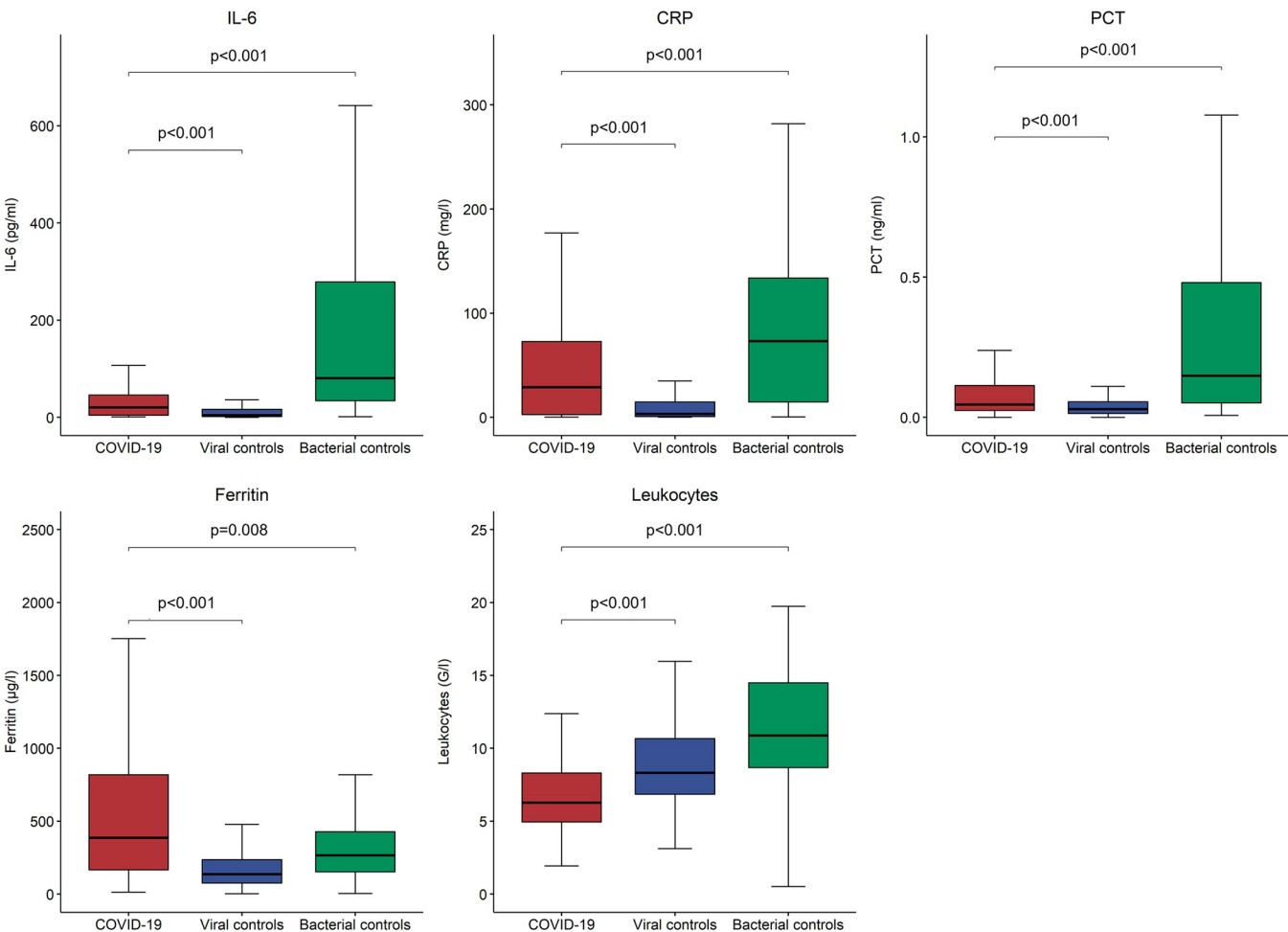

**Fig 2. Distribution of inflammatory biomarkers in COVID-19 and controls at ED presentation.** P-values were calculated using the Mann-Whitney-U test; COVID-19 = coronavirus disease 2019, IL-6 = interleukin-6, CRP = c-reactive protein, PCT = procalcitonin, ED = emergency department.

again <0.001 for all comparisons. Ferritin levels were highest in COVID-19 cases compared to viral controls (387 µg/l [IQR 164–823] vs. 137 µg/l [IQR 76–238], p<0.001) and bacterial controls (387 µg/l [IQR 164–823] vs. 266 µg/l [IQR 153–435], p = 0.008). Leukocytes were lowest in COVID-19 cases compared to viral controls (6.27 G/l [IQR 4.95–8.34] vs. 8.34 G/l [IQR 6.85–10.70], p<0.001) and bacterial controls (6.27 G/l [IQR 4.95–8.34] vs. 10.89 G/l [IQR 8.65–14.55], p<0.001). All five investigational inflammatory biomarkers correlated with the severity of the disease in COVID-19 but, at least partly, also in patients with other respiratory infections (S1 Fig).

## 3.3. Utility of inflammatory biomarkers to predict need for hospitalisation

Table 2 shows baseline characteristics of COVID-19 cases and controls, stratified by subsequent hospitalisation after ED presentation. Age, cardiac disease, and higher respiratory rate were identified as risk factors, regardless of the final diagnosis. In contrast, cough and dyspnea could not be identified as clear risk factors. As displayed in Fig 3, all inflammatory biomarkers in COVID-19 patients were significantly higher in hospitalised patients than in non-hospitalised patients, e.g., CRP 59.3 mg/l (IQR 31.5–126.9) vs. 2.3 mg/l (IQR 0.9–11.1), p<0.001. In

**Table 2. Baseline characteristics in hospitalised and non-hospitalised cases in COVID-19 and controls.**

| Measures | COVID-19 | | p-value | Viral controls | | p-value | Bacterial controls | | p-value |
|---|---|---|---|---|---|---|---|---|---|
| | n = 191 | | | n = 227 | | | n = 96 | | |
| | No hospitalisation | Hospitalisation | | No hospitalisation | Hospitalisation | | No hospitalisation | Hospitalisation | |
| | n = 76 | n = 115 | | n = 169 | n = 58 | | n = 13 | n = 83 | |
| Demographics | | | | | | | | | |
| Age—years | 46 [36–57] | 62 [52–75] | <0.001 | 48 [34–61] | 65 [52–75] | <0.001 | 65 [37–69] | 73 [59–80] | 0.027 |
| Female | 40 (53) | 44 (38) | 0.050 | 82 (49) | 23 (40) | 0.243 | 3 (23) | 34 (41) | 0.218 |
| Comorbidities—no (%) | | | | | | | | | |
| Cardiac disease[a] | 6 (8) | 32 (28) | 0.001 | 23 (14) | 28 (48) | <0.001 | 2 (15) | 39 (47) | 0.032 |
| • Valvular cardiopathy | 0 (0) | 8 (7) | 0.019 | 3 (2) | 6 (10) | 0.004 | 0 (0) | 7 (8) | 0.277 |
| • Coronary artery disease | 2 (3) | 19 (17) | 0.003 | 10 (6) | 16 (28) | <0.001 | 0 (0) | 17 (20) | 0.072 |
| • Prior myocardial infarction | 1 (1) | 8 (7) | 0.072 | 5 (3) | 7 (12) | 0.007 | 0 (0) | 10 (12) | 0.186 |
| • Atrial fibrillation | 1 (1) | 8 (7) | 0.072 | 4 (2) | 6 (10) | 0.011 | 1 (8) | 22 (27) | 0.139 |
| Hypertension | 15 (20) | 66 (57) | <0.001 | 56 (33) | 31 (53) | 0.006 | 5 (38) | 50 (60) | 0.140 |
| Overweight | 12 (16) | 62 (54) | <0.001 | 43 (25) | 26 (45) | 0.006 | 3 (23) | 19 (23) | 0.988 |
| Diabetes | 3 (4) | 31 (27) | <0.001 | 13 (8) | 14 (24) | 0.001 | 3 (23) | 21 (25) | 0.863 |
| Ever smoker | 19 (25) | 39 (34) | 0.190 | 65 (38) | 40 (69) | <0.001 | 8 (62) | 46 (55) | 0.679 |
| Pneumopathy[b] | 14 (18) | 23 (20) | 0.787 | 55 (33) | 33 (57) | 0.001 | 6 (46) | 33 (40) | 0.662 |
| • Asthma | 13 (17) | 12 (10) | 0.181 | 35 (21) | 6 (10) | 0.077 | 4 (31) | 9 (11) | 0.051 |
| • COPD | 0 (0) | 9 (8) | 0.012 | 15 (9) | 22 (38) | <0.001 | 1 (8) | 20 (24) | 0.183 |
| Hepatopathy | 5 (7) | 9 (8) | 0.746 | 11 (7) | 12 (21) | 0.002 | 2 (15) | 12 (14) | 0.930 |
| CKD | 0 (0) | 26 (23) | <0.001 | 6 (4) | 8 (14) | 0.005 | 1 (8) | 24 (29) | 0.105 |
| Stroke | 2 (3) | 8 (7) | 0.189 | 2 (1) | 7 (12) | <0.001 | 0 (0) | 10 (12) | 0.186 |
| Cancer | 4 (5) | 13 (11) | 0.151 | 5 (3) | 7 (12) | 0.007 | 2 (15) | 16 (19) | 0.738 |
| Immunodeficiency | 3 (4) | 8 (7) | 0.382 | 8 (5) | 3 (5) | 0.893 | 1 (8) | 13 (16) | 0.449 |
| Symptoms at ED—(%) | | | | | | | | | |
| Symptom duration before ED—days | 7 [2–12] | 7 [3–10] | 0.569 | 5 [2–10] | 4 [2–10] | 0.724 | 6 [4–10] | 3 [2–7] | 0.017 |
| Cough | 50 (66) | 76 (66) | 0.966 | 137 (81) | 45 (78) | 0.566 | 12 (92) | 48 (58) | 0.017 |
| Dyspnea | 31 (41) | 50 (43) | 0.713 | 96 (57) | 40 (69) | 0.103 | 3 (23) | 46 (55) | 0.030 |
| Vital signs at ED | | | | | | | | | |
| Systolic BP—mmHg | 135 [123–151] | 134 [120–148] | 0.645 | 142 [127–155] | 140 [120–160] | 0.939 | 127 [119–143] | 133 [120–153] | 0.516 |
| Diastolic BP—mmHg | 83 [74–90] | 80 [70–90] | 0.127 | 82 [74–89] | 82 [71–89] | 0.801 | 79 [74–94] | 80 [68–86] | 0.516 |
| Heart rate—/min | 87 [80–100] | 90 [80–105] | 0.201 | 86 [75–100] | 92 [78–102] | 0.142 | 94 [84–100] | 98 [77–110] | 0.825 |
| Blood oxygen saturation—% | 98 [97–99] | 95 [93–97] | <0.001 | 98 [97–98] | 96 [94–98] | <0.001 | 96 [95–99] | 95 [92–97] | 0.077 |
| Respiratory rate—/min | 17 [15–21] | 23 [16–25] | <0.001 | 17 [15–20] | 20 [16–25] | <0.001 | 17 [15–20] | 24 [20–28] | <0.001 |
| Temperature - ˚C | 37.0 [36.6–37.4] | 37.3 [36.8–38.2] | 0.003 | 36.8 [36.5–37.2] | 37.0 [36.6–37.9] | 0.041 | 37.1 [36.7–38.1] | 37.4 [37.0–38.5] | 0.143 |
| Laboratory parameters at ED | | | | | | | | | |
| IL-6—pg/ml | 4.34 [2.09–10.74] | 40.90 [20.92–64.17] | <0.001 | 3.10 [1.81–9.67] | 21.93 [9.86–82.68] | <0.001 | 28.34 [6.47–125.04] | 94.83 [35.26–356.17] | 0.022 |
| CRP—mg/l | 2.3 [0.9–11.1] | 59.3 [31.5–126.9] | <0.001 | 2.3 [0.7–9.4] | 13.9 [3.1–49.8] | <0.001 | 15.1 [5.4–46.6] | 85.3 [24.0–142.2] | 0.011 |
| PCT—ng/ml | 0.026 [0.013–0.045] | 0.082 [0.042–0.193] | <0.001 | 0.028 [0.014–0.047] | 0.048 [0.015–0.102] | 0.006 | 0.066 [0.037–0.101] | 0.170 [0.052–0.671] | 0.085 |
| Ferritin - µg/l | 193 [95–361] | 672 [324–1258] | <0.001 | 126 [78–226] | 165 [64–301] | 0.240 | 177 [124–293] | 275 [156–477] | 0.198 |

(Continued)

**Table 2.** (Continued)

| Measures | COVID-19 | | p-value | Viral controls | | p-value | Bacterial controls | | p-value |
|---|---|---|---|---|---|---|---|---|---|
| | n = 191 | | | n = 227 | | | n = 96 | | |
| | No hospitalisation | Hospitalisation | | No hospitalisation | Hospitalisation | | No hospitalisation | Hospitalisation | |
| | n = 76 | n = 115 | | n = 169 | n = 58 | | n = 13 | n = 83 | |
| Leukocytes—G/l | 5.83 [4.59–7.15] | 6.67 [5.25–8.93] | 0.012 | 8.21 [6.61–10.38] | 9.16 [7.12–12.29] | 0.006 | 9.62 [8.56–11.32] | 11.14 [8.73–14.73] | 0.349 |

[a] cardiac disease includes valvular cardiopathy, coronary artery disease, prior myocardial infarction, and atrial fibrillation

[b] pneumopathy includes asthma and COPD.

p-values for comparison of clinical characteristics regarding hospitalisation after ED visit, continuous variables were calculated using the Mann-Whitney-U test, and categorical variables using the Pearson χ2 test or Fisher's exact test, as appropriate. Values are numbers (percentages) or median [interquartile range]

COVID-19 = coronavirus disease 2019, COPD = chronic obstructive pulmonary disease, CKD = chronic kidney disease, ED = emergency department, BP = blood pressure, IL-6 = interleukin-6, CRP = c-reactive protein, PCT = procalcitonin.

both control groups, inflammatory biomarkers were numerically higher in hospitalised patients than in non-hospitalised patients with significant differences in both groups for IL-6 and CRP. E.g., in viral controls, CRP was 13.9 mg/l (IQR 3.1–49.8) vs. 2.3 mg/l (IQR 0.7–9.4), p<0.001, whereas in bacterial controls CRP was 85.3 mg/l (IQR 24.0–142.2) vs. 15.1 mg/l (IQR 5.4–46.6), p = 0.011. **Fig 4** displays the ROC for the outcome hospitalisation for inflammatory biomarkers in all three groups. IL-6 and CRP had the largest AUC regardless of the diagnosis. In COVID-19, IL-6 and CRP showed very high discriminative utility with an AUC of 0.899 (95%CI 0.850–0.948) for IL-6 and 0.922 (95%CI 0.879–0.946) for CRP, respectively. Similarly, in the logistic regression model for the primary outcome of hospitalisation after ED presentation for all inflammatory biomarkers, displayed in **Table 3**, IL-6 and CRP showed the highest predictive value in univariable analyses (IL-6 OR 31.836 [95%CI, 11.310–89.609], p<0.001, CRP OR 14.528 [95%CI, 6.602–31.971], p<0.001) for COVID-19. Additionally, in the multivariable model with all inflammatory biomarkers combined, at least one of these two biomarkers remained an independent predictor regardless of the diagnosis.

## 3.4. Utility of inflammatory biomarkers to predict ICU admission or death at 30 days

Distribution of inflammatory parameters according to the secondary composite outcome of ICU admission or death at 30 days is displayed in **Fig 5**. In COVID-19 patients with a secondary outcome, all investigated inflammatory biomarkers were significantly higher compared to event-free survivors, e.g., CRP 112.1 mg/l (IQR 47.6–162.0) vs. 15.1 mg/l (IQR 1.6–46.5), p<0.001 (**S3 Table**). In controls, IL-6 and CRP were the only biomarkers to systematically show significant differences between patients with and without secondary outcomes. Of note, ferritin did not differ between patients with and without secondary outcomes in both control groups. **Fig 6** displays the ROC for the secondary composite outcome. IL-6 and CRP showed good discrimination in COVID-19, with an AUC of 0.794 (95%CI, 0.694–0.894) for IL-6 and 0.807 (95%CI, 0.721–0.893) for CRP. **Fig 7** shows the event curve for IL-6 and CRP in COVID-19 for the secondary outcome. Incidence of the secondary outcome was 23% in COVID-19 patients with high IL-6 levels (above the median of 20.77 pg/ml) vs. 5% in patients with low IL-6 levels (log-rank p = 0.002, HR 4.62 [95%CI, 1.55–13.73], p = 0.006). Patients with high CRP levels (above the median of 28.9 mg/l) show an event rate of 37% vs. 5% in patients with low CRP levels, (log-rank p<0.001, HR 7.88 [95%CI, 3.08–20.18], p<0.001).

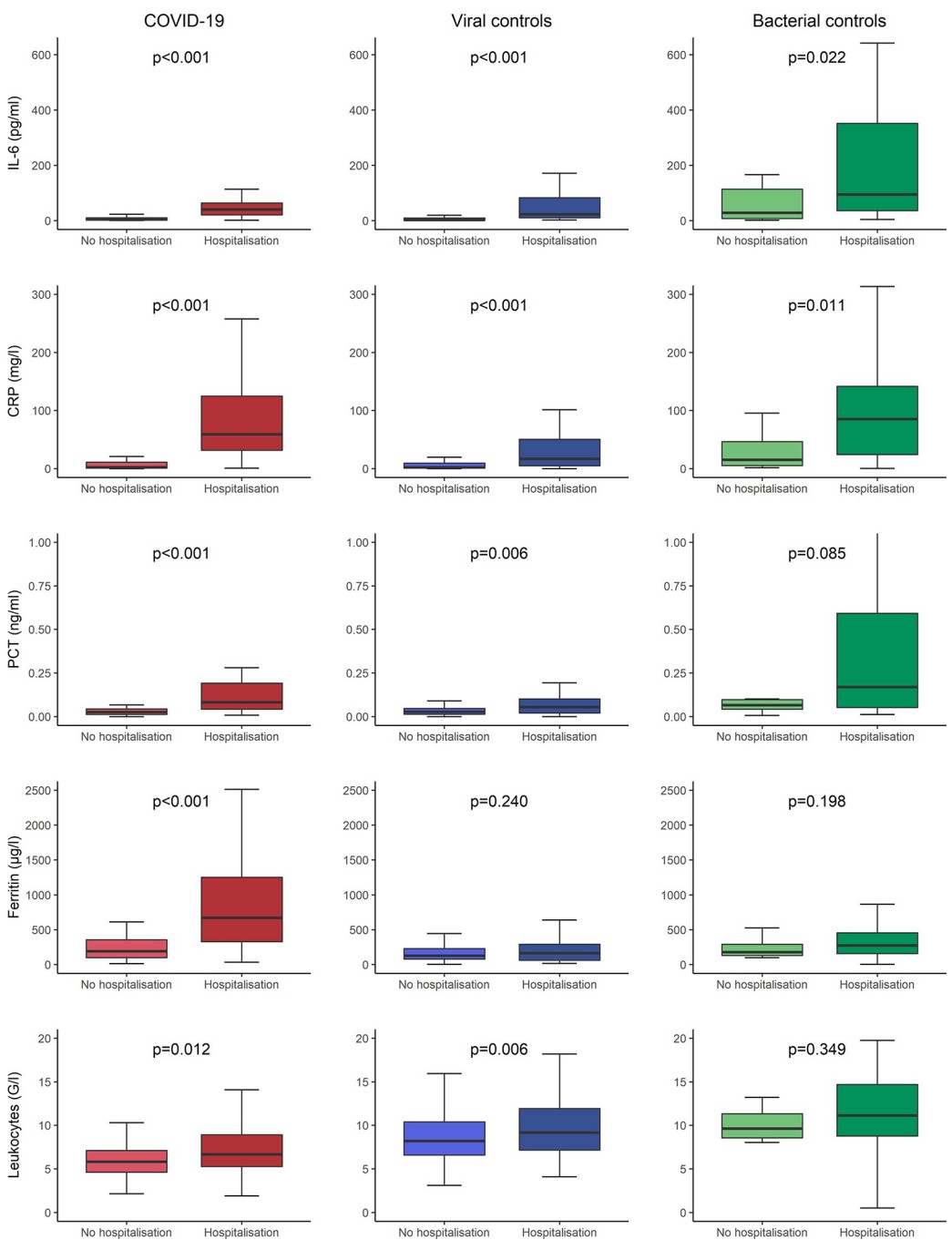

**Fig 3. Distribution of inflammatory biomarkers in COVID-19 and controls regarding the primary outcome.** Primary outcome was the need for hospitalisation at ED presentation; P-values were calculated using the Mann-Whitney-U test; COVID-19 = coronavirus disease 2019, IL-6 = interleukin-6, CRP = c-reactive protein, PCT = procalcitonin, ED = emergency department.

**Table 4** shows Cox regression analysis of the secondary composite outcome of ICU admission or death at 30 days for all inflammatory biomarkers. In COVID-19, IL-6 and CRP showed the highest prognostic value in the univariable model. In the multivariable model with all inflammatory biomarkers combined, at least one of these two biomarkers remained in the final model as an independent predictor regardless of the diagnosis.

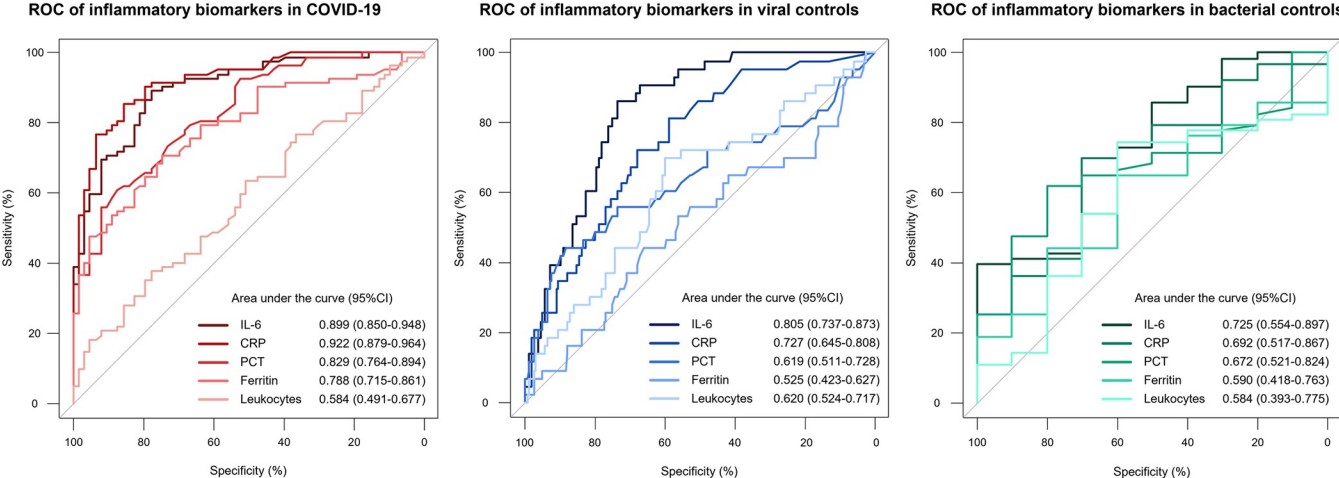

**Fig 4. Discriminative performance of inflammatory biomarkers regarding the primary outcome in COVID-19 and controls.** ROC for the primary outcome of hospitalisation at ED presentation in COVID-19 and controls; ROC = receiver operating characteristic curves, COVID-19 = coronavirus disease 2019, IL-6 = interleukin-6, CRP = c-reactive protein, PCT = procalcitonin, ED = emergency department.

## 4. Discussion

### 4.1. Findings

In this observational prospective single-centre cohort study of cases presenting with suspected SARS-CoV-2 infection to the ED of the University Hospital in Basel, Switzerland, we explore and directly compare the predictive value of inflammatory biomarkers between patients with confirmed COVID-19 and patients with respiratory infections from other cause. We report six major findings.

**First**, levels of inflammatory biomarkers differ significantly between cases with COVID-19, viral controls, and bacterial controls. Bacterial controls have the highest levels of IL-6, CRP,

**Table 3. Binary logistic regression model for the outcome of hospitalisation.**

| Measures | COVID-19 | | | | Viral controls | | | | Bacterial controls | | | |
|---|---|---|---|---|---|---|---|---|---|---|---|---|
| | *n* = 191 | | | | *n* = 227 | | | | *n* = 96 | | | |
| | Univariable | | Multivariable | | Univariable | | Multivariable | | Univariable | | Multivariable | |
| | OR (95% CI) | *p-value* | OR (95% CI) | *p-value* | OR (95% CI) | *p-value* | OR (95% CI) | *p-value* | OR (95% CI) | *p-value* | OR (95% CI) | *p-value* |
| IL-6 | 31.836 (11.31–89.609) | <0.001 | 3.931 (0.877–17.627) | 0.074 | 5.432 (3.004–9.822) | <0.001 | 5.432 (3.004–9.822) | <0.001 | 4.170 (1.348–12.902) | 0.013 | 4.170 (1.348–12.902) | 0.013 |
| CRP | 14.528 (6.602–31.971) | <0.001 | 6.763 (2.261–20.227) | <0.001 | 3.229 (1.978–5.271) | <0.001 | - | | 2.628 (1.003–6.885) | 0.049 | - | |
| PCT | 25.841 (7.697–86.753) | <0.001 | - | | 2.153 (1.226–3.784) | 0.008 | - | | 3.183 (0.921–11.008) | 0.067 | - | |
| Ferritin | 12.043 (4.709–30.796) | <0.001 | - | | 0.989 (0.455–2.151) | 0.978 | - | | 1.286 (0.316–5.231) | 0.726 | - | |
| Leukocytes | 5.984 (0.978–36.611) | 0.053 | - | | 17.835 (1.605–198.189) | 0.019 | - | | 1.168 (0.085–16.007) | 0.907 | - | |

p-values for comparison of OR were calculated using Fisher's exact test

values were logarithmized to approach a normal distribution, values for the multivariable model were selected using a backwards selection process

OR = odds ratio, CI = confidence interval, IL-6 = interleukin-6, CRP = c-reactive protein, PCT = procalcitonin.

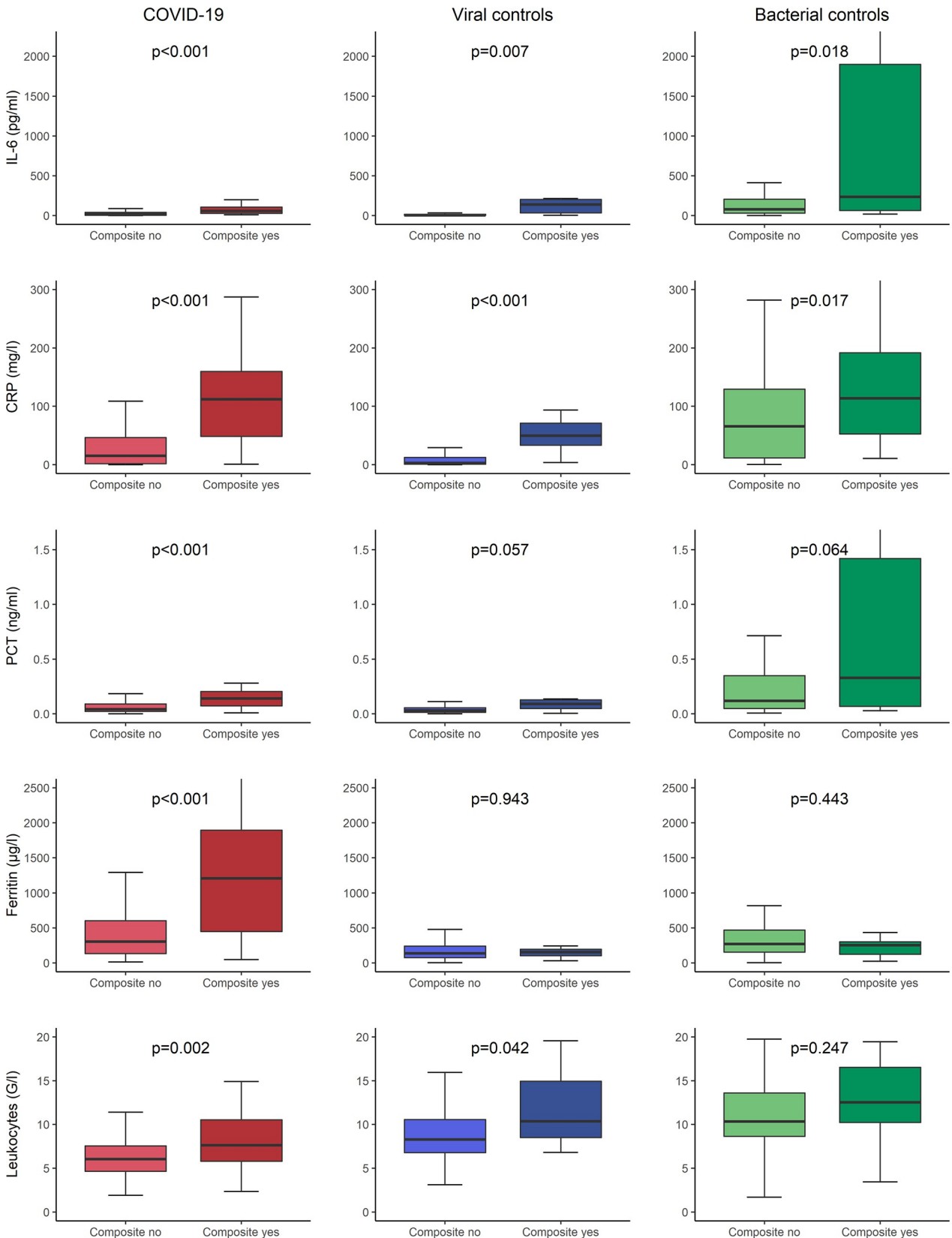

**Fig 5. Distribution of inflammatory biomarkers in COVID-19 and controls regarding secondary outcome.** Secondary outcome was the composite of ICU admission or death at 30 days; P-values were calculated using the Mann-Whitney-U test; COVID-19 = coronavirus disease 2019, IL-6 = interleukin-6, CRP = c-reactive protein, PCT = procalcitonin, ICU = intensive care unit.

PCT and leukocytes. In contrast, patients with COVID-19 show the highest levels of ferritin and the lowest levels of leukocytes among the three diagnoses. Viral controls have the lowest inflammatory levels of biomarkers overall. This finding corroborates the results from other studies reporting low leukocyte levels in COVID-19 [10, 46, 49, 50]. However, in contrast to most previous studies, our analyses allow to interpret such findings in direct comparison with other respiratory infections. Whether inflammatory reactions in COVID-19 are in general truly more pronounced than in other viral respiratory infections but less than in bacterial pneumonia remains speculative and needs to be addressed in future studies.

**Second,** inflammatory biomarkers are strongly associated with the severity of the disease for all diagnostic entities, but particularly for COVID-19. This finding is in line with previous observations from earlier studies about COVID-19 [3, 5, 6, 8, 9, 14]. For example, Liu et al. came to similar conclusions, as they found IL-6 and CRP to be independent predictors of disease severity in COVID-19 patients. However, in their study only three biomarkers were compared [3].

**Third**, levels of inflammatory biomarkers, especially IL-6 and CRP, show high discriminative accuracy to predict the need for hospitalisation in patients with COVID-19. This finding confirms early observations suggesting a predictive value of IL-6 and CRP in patients with COVID-19 [1, 3, 8, 10].

**Fourth,** when assessed in patients with viral or bacterial respiratory infections, the utility of inflammatory biomarkers to predict the need for hospitalisation is moderate and in general lower than in COVID-19.

**Fifth**, IL-6 and CRP are the two best performing inflammatory biomarkers to predict ICU admission or death at 30 days in patients with COVID-19. When assessed in a multivariable model, IL-6 provides the best predictive performance out of the investigational five biomarkers. These findings are in line with previous studies investigating the prognostic utility of biomarkers in COVID-19 patients [8, 9].

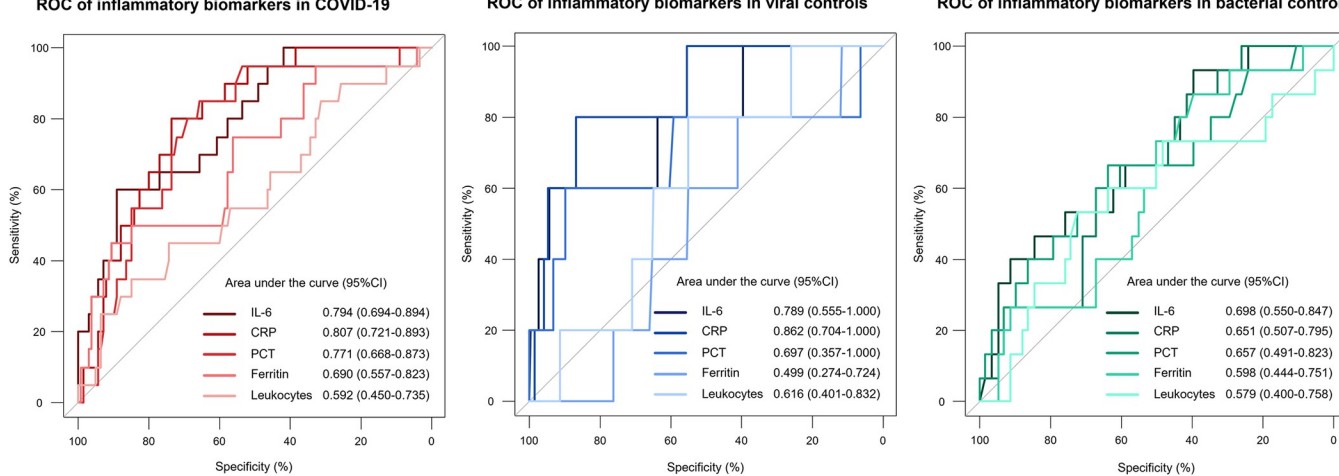

**Fig 6. Discriminative performance of inflammatory biomarkers regarding the secondary outcome in COVID-19 and controls.** ROC for the secondary outcome of ICU admission or death at 30 days; ROC = receiver operating characteristic curves, COVID-19 = coronavirus disease 2019, IL-6 = interleukin-6, CRP = c-reactive protein, PCT = procalcitonin, ICU = intensive care unit.

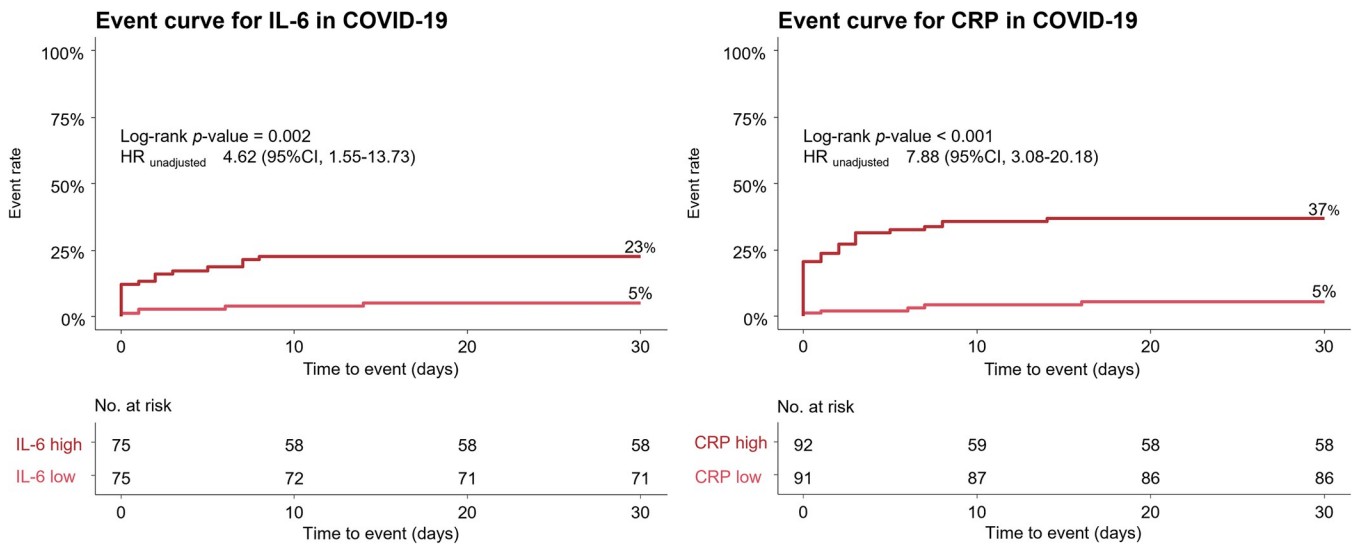

**Fig 7. Event curves of COVID-19 cases for the secondary outcome stratified by IL-6 and CRP.** Event curves for the secondary outcome of the composite of ICU-admission or death at 30 days. The respective medians served as cut-off values for IL-6 (left) and CRP (right). Numbers at risk are displayed at the bottom of the figure; P-values were calculated using the log-rank test; IL-6 = interleukin-6, COVID-19 = coronavirus disease 2019, CRP = c-reactive protein, HR = hazard ratio.

**Last**, when assessed in viral controls or bacterial controls, inflammatory biomarkers provide moderate utility in both control groups, however, the performance is lower than in COVID-19. As in COVID-19, IL-6 and CRP show the best predictive utility.

## 4.2. Strengths and limitations

This study has several strengths and limitations.

One major strength of this study is its prospective design and the consecutive recruitment of unselected cases. To our knowledge, there is still a systematic lack of prospective cohort

**Table 4. Cox regression model for the outcome of the composite endpoint.**

| Measures | COVID-19 | | | | Viral controls | | | | Bacterial controls | | | |
|---|---|---|---|---|---|---|---|---|---|---|---|---|
| | *n* = 191 | | | | *n* = 227 | | | | *n* = 96 | | | |
| | Univariable | | Multivariable | | Univariable | | Multivariable | | Univariable | | Multivariable | |
| | HR (95% CI) | *p-value* | HR (95% CI) | *p-value* | HR (95% CI) | *p-value* | HR (95% CI) | *p-value* | HR (95% CI) | *p-value* | HR (95% CI) | *p-value* |
| IL-6 | 6.602 (3.113–14.003) | <0.001 | 6.602 (3.113–14.003) | <0.001 | 3.630 (1.602–8.227) | 0.002 | - | - | 2.277 (1.200–4.321) | 0.012 | 2.101 (1.042–4.239) | 0.038 |
| CRP | 5.519 (2.187–13.932) | <0.001 | - | - | 7.651 (1.562–37.48) | 0.012 | 7.651 (1.562–37.480) | 0.012 | 2.484 (0.914–6.754) | 0.075 | 3.091 (0.839–11.393) | 0.090 |
| PCT | 2.796 (1.476–5.297) | 0.002 | - | - | 3.445 (1.414–8.394) | 0.006 | - | - | 1.862 (1.064–3.258) | 0.029 | - | - |
| Ferritin | 4.397 (1.632–11.845) | 0.003 | - | - | 0.927 (0.126–6.811) | 0.941 | - | - | 0.558 (0.215–1.449) | 0.231 | 0.290 (0.082–1.022) | 0.054 |
| Leukocytes | 5.066 (0.834–30.771) | 0.078 | - | - | 10.899 (0.027–4377.491) | 0.435 | - | - | 0.710 (0.099–5.116) | 0.734 | - | - |

p-values for comparison of HR were calculated using the log-rank test

values were logarithmized to approach a normal distribution, values for the multivariable model were selected using a backwards selection process

HR = hazard ratio, CI = confidence interval, IL-6 = interleukin-6, CRP = c-reactive protein, PCT = procalcitonin.

studies assessing clinical characteristics and laboratory parameters of COVID-19 during the ongoing pandemic. This approach harbours the advantages of minimizing a potential recall bias and allow for more complete data collection.

Similarly, the presence of adequate control groups as provided in this study represent another strength. This allows to directly compare the clinical utility of the investigational inflammatory biomarkers in cases with COVID-19 and in control groups of cases with respiratory infections other than COVID-19 but with similar symptoms recruited at the same time period. Unfortunately, in most studies on COVID-19, clinical signs and biomarkers are exclusively explored in an isolated fashion, focusing only on COVID-19. However, the direct link to comparable clinical settings such as other viral respiratory infections or pneumonia is largely missing. The presence of adequate control groups, however, is mandatory to compare the clinical utility of clinical signs and biomarkers and test whether they are COVID-19-specific or generalizable to all cases presenting to the ED with acute respiratory infections.

There are, however, also several limitations.

**First**, only 191 cases with COVID-19 and 323 controls were included in this study. Overall, 75 combined events for the secondary outcome were recorded, which was mainly driven by events in COVID-19 patients and bacterial pneumonia. While this allows to assess biomarker signatures in a descriptive fashion, statistical power may be insufficient for extensive multivariable and subgroup analyses.

**Second**, despite our efforts to differentiate viral from bacterial respiratory infections based on clinical examinations and radiological findings, we cannot guarantee that a small proportion of patients were misclassified due to the missing routine distinction of pathogens.

**Third**, this study contains numerous comparisons with no a-priori adjustment for multiple testing. Accordingly, p-values must be interpreted with caution. Similarly, due to the rather small sample size, no inter-group adjustment for potential confounders (e.g., age, comorbidities) was applied.

**Fourth**, despite the prospective study design, some inflammatory biomarkers were still missing in some subjects. This was mostly true for IL-6 and PCT, as these two biomarkers were measured at an external facility and therefore needed an additional blood serum sample stored in the dedicated biobank.

**Fifth**, despite our efforts to minimize the error of misclassification by carefully analysing available SARS-CoV-2 PCR test results, there is still the possibility of some false negatives in the respective control groups.

**Sixth,** treating physicians were blinded to the results of IL-6 and PCT, but not CRP, ferritin, and leukocytes, as they were part of the clinical routine panel. Therefore, these biomarkers might have played some role in the management decision at the time of ED triage and could have led to performance bias.

## 5. Conclusion

In patients with COVID-19 and other respiratory infections, inflammatory biomarkers harbour strong prognostic information, particularly IL-6 and CRP. Their routine use might further improve early management decision.

## Supporting information

**S1 Fig. Distribution of inflammatory biomarkers in COVID-19 and controls regarding disease severity.** Disease severity is categorized in four categories; outpatients, normal ward, ICU admission, and death at 30 days; P-values were calculated using the Kruskal-Wallis test; COVID-19 = coronavirus disease 2019, IL-6 = interleukin-6, CRP = c-reactive protein,

PCT = procalcitonin, ICU = intensive care unit.
(TIF)

**S1 Table. STROBE statement.**
(DOCX)

**S2 Table. Missing values.**
(DOCX)

**S3 Table. Inflammatory biomarkers regarding the secondary outcome in COVID-19 and controls.**
(DOCX)

## Acknowledgments

We thank all the physicians and caregivers at the emergency department, ward, and intensive care unit for their help in this study during this difficult time.

## Author Contributions

**Conceptualization:** Maurin Lampart, Stefan Osswald, Gabriela M. Kuster, Raphael Twerenbold.

**Data curation:** Maurin Lampart, Núria Zellweger, Raphael Twerenbold.

**Formal analysis:** Maurin Lampart.

**Funding acquisition:** Maurin Lampart, Raphael Twerenbold.

**Investigation:** Maurin Lampart, Núria Zellweger, Raphael Twerenbold.

**Methodology:** Maurin Lampart, Raphael Twerenbold.

**Project administration:** Maurin Lampart, Stefano Bassetti, Sarah Tschudin-Sutter, Katharina M. Rentsch, Martin Siegemund, Roland Bingisser, Stefan Osswald, Gabriela M. Kuster, Raphael Twerenbold.

**Resources:** Maurin Lampart, Katharina M. Rentsch, Martin Siegemund, Roland Bingisser, Stefan Osswald, Gabriela M. Kuster, Raphael Twerenbold.

**Software:** Maurin Lampart, Raphael Twerenbold.

**Supervision:** Maurin Lampart, Raphael Twerenbold.

**Validation:** Maurin Lampart, Raphael Twerenbold.

**Visualization:** Maurin Lampart, Raphael Twerenbold.

**Writing – original draft:** Maurin Lampart, Raphael Twerenbold.

**Writing – review & editing:** Maurin Lampart, Núria Zellweger, Roland Bingisser, Gabriela M. Kuster, Raphael Twerenbold.

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
