## [Decision Letter · Decision Letter 0]

25 Jan 2022

PONE-D-21-39440Clinical utility of inflammatory biomarkers in COVID-19 in direct comparison to other respiratory infections - A prospective cohort studyPLOS ONE

Dear Dr. Lampart,

Thank you for submitting your manuscript to PLOS ONE. After careful consideration, we feel that it has merit but does not fully meet PLOS ONE’s publication criteria as it currently stands. Therefore, we invite you to submit a revised version of the manuscript that addresses the points raised during the review process.

ACADEMIC EDITOR: The study by Lampart and coauthors is overall interesting and informative. However major revisions are required, as suggested by the reviewers, before it can be considered for publication. 

We look forward to receiving your revised manuscript.

Kind regards,

Paola Faverio

Academic Editor

PLOS ONE

Journal Requirements:

[I have read the journal's policy and the authors of this manuscript have the following competing interests: RT reports research support from the Swiss National Science Foundation (Grant No P300PB_167803), the Swiss Heart Foundation, the Swiss Society of Cardiology, the

Cardiovascular Research Foundation Basel, the University of Basel and the University Hospital Basel and speaker honoraria/consulting honoraria from Abbott, Amgen, Astra Zeneca, Roche, Siemens,

Singulex, and Thermo Scientific BRAHMS. GK reports research support from the Swiss National

Science Foundation (Grant No IZCOZ0_189877) and the Cardiovascular Research Foundation Basel,

that are unrelated to this work, and consultant fees from Janssen. Authors not named

here have disclosed no competing interests.]

4. Thank you for submitting the above manuscript to PLOS ONE. During our internal evaluation of the manuscript, we found significant text overlap between your submission and the following previously published works, some of which you are an author.

Lampart M, Rüegg M, Jauslin AS, Simon NR, Zellweger N, Eken C, Tschudin-Sutter S, Bassetti S, Rentsch KM, Siegemund M, Bingisser R, Nickel CH, Osswald S, Kuster GM, Twerenbold R. Direct Comparison of Clinical Characteristics, Outcomes, and Risk Prediction in Patients with COVID-19 and Controls—A Prospective Cohort Study. Journal of Clinical Medicine. 2021; 10(12):2672. https://doi.org/10.3390/jcm10122672

Please revise the manuscript to rephrase the duplicated text, cite your sources, and provide details as to how the current manuscript advances on previous work. Please note that further consideration is dependent on the submission of a manuscript that addresses these concerns about the overlap in text with published work.

Reviewers' comments:

Reviewer's Responses to Questions

**Comments to the Author**

1. Is the manuscript technically sound, and do the data support the conclusions?

Reviewer #1: Yes

Reviewer #2: Yes

2. Has the statistical analysis been performed appropriately and rigorously? 

Reviewer #1: Yes

Reviewer #2: Yes

3. Have the authors made all data underlying the findings in their manuscript fully available?

Reviewer #1: Yes

Reviewer #2: Yes

4. Is the manuscript presented in an intelligible fashion and written in standard English?

Reviewer #1: Yes

Reviewer #2: Yes

5. Review Comments to the Author

Reviewer #1: This is an interesting study, one of the first trying to compare inflammatory makers in covid and other respiratory infections. However, some aspects of the study, especially the classification of other respiratory infections, could be better explained.

INTRODUCTION

The immunological mechanisms underlying the increase / decrease of markers are explained in a simplistic way. I suggest expanding the introduction by adding more information (e.g. adding the normal values of the cited markers, explaining for each marker the (presumed) mechanism of increase/decreased both in covid / bacterial and viral infections.

For example, about leukocyte levels:

I suggest adding: “Leukocytosis is defined by an increase in the WBC count of more than 11,000 cells/microL.”

Line 92: “Leukocyte levels often increase during infections due to the activation of the immune

system.” I suggest to better explain the concepts of neutrophilic leukocytosis in bacterial infection and lymphocytosis/lymphopenia in viral infection, so for example:

Leukocyte levels often increase during infections, due to the release of several molecules, as growth or survival factors, adhesion molecules and various cytokines released during activation of immune system. Most bacterial infections are associated with neutrophilic leukocytosis. Neutrophilia occurs from both upregulated bone marrow production and the release of neutrophils from the endothelium. Generally, most viruses lead to relative lymphocytosis, while only a few viruses causing could result in lymphopenia, such as SARS-COV-2.

Line 96-97: “as SARS CoV-2 binds to the angiotensin converting enzyme 2 (ACE2), which is located on most lymphocytes” This sentence is not enough specific to explain the mechanism of lymphopenia. I suggest to better specify the mechanisms of lymphopenia in covid-19 (for example, referring to Jafarzadeh A, et al. Lymphopenia an important immunological abnormality in patients with COVID-19: Possible mechanisms. Scand J Immunol. 2021)

MATERIALS AND METHODS

Line 106: “included unselected patients aged 18 years and older 106 presenting with clinically suspected or confirmed SARS-CoV-2 infection”. Was the enrollment of patients consecutive? Please specify.

Line 126: Why do you choose “leukocytes” and you don’t differentiate between neutrophils and lymphocytes? Please specify.

Line 132-133: Why treating physicians were not blinded for the other markers? Please specify.

Line 158: Please add the reference to S1 (flowchart of the study). Is it possible to move the flowchart of the study from the supplementary material to the main text? the flow chart is well constructed and helps the reader to follow the text.

Line 158-161: How five trained physician differentiate between viral and bacterial pneumonia? Did you use specific criteria (e.g. bacterium isolation on sputum sample, urinary antigen positivity for pneumococcus, virus isolation on multiplex PCR, lobar or interstitial pneumonia on chest x-ray)? Or the decision was only “clinical”? Please specify. This is a crucial point of the study design.

Is it possible to have some information relating to the identification of bacteria or other viruses?

RESULTS

Table 1: were coronary artery disease, prior myocardial infarction, atrial fibrillation and hypertension included in “cardiac disease”? please specify.

In table 1 we can see that some patients have immunodeficiency (even if the cause of immunodeficiency is not specified). Do you think the immunodeficiency could altered the inflammatory response (and therefore consequently also the values of the analyzed markers)? Please explain the choice to include immunodeficiency in the analyses.

Line 233-234. As you described first the hospitalized group, I suggest inverting the example because (e.g., CRP 2.3 mg/l (IQR 0.9-234 11.1) refers to non-hospitalized patients while 59.3 mg/l (IQR 31.5-126.9) refers to hospitalized patient. p<0.001. The same for line 237 and 238.

DISCUSSION

Discussion clearly reports the results without comparing them with other studies. Please add, if possible, more comparison with other studies such as those mentioned in the introduction.

Line 325-327: “This finding suggests that inflammatory reactions in COVID-19 are in general more pronounced than in other viral respiratory infections but less than in bacterial pneumonia”. This conclusion is speculative: the fact that the inflammatory markers are different in the three cases does not mean that the inflammatory mechanism in covid 19 is less pronounced. Please rephrase.

Line 383: please add to the limit session also the risk of misclassification of bacterial and viral pneumoniae (the method of diagnosis is not clear).

Reviewer #2: Thank you for the opportunity to review this article

The work of Lampart et al. is interesting and analyzed the role of inflammatory factors in COVID-19 and other respiratory infections. The article outlines some clinically important aspects.

I have some observations:

In Materials and methods, line 106: “…infection to the ED of….” What does ED mean? Please put it in full

In Materials and Methods how were other viral and bacterial infections (viral and bacterial controls) diagnosed? Please specify diagnostic methods and viral and bacterial diagnosis.

The bacterial control was significantly older and with more comorbidities than the others group. In the Author’s opinion, could this have contributed to the higher values of the inflammatory indices?

6. PLOS authors have the option to publish the peer review history of their article (what does this mean?). If published, this will include your full peer review and any attached files.

Reviewer #1: No

Reviewer #2: No

---

## [Author Response · Author response to Decision Letter 0]

25 Mar 2022

Academic Editor

Answer: We have carefully checked the style requirements for PLOS One and named the tables and figures accordingly.

[I have read the journal's policy and the authors of this manuscript have the following competing interests: RT reports research support from the Swiss National Science Foundation (Grant No P300PB_167803), the Swiss Heart Foundation, the Swiss Society of Cardiology, the Cardiovascular Research Foundation Basel, the University of Basel and the University Hospital Basel and speaker honoraria/consulting honoraria from Abbott, Amgen, Astra Zeneca, Roche, Siemens, Singulex, and Thermo Scientific BRAHMS. GK reports research support from the Swiss National Science Foundation (Grant No IZCOZ0_189877) and the Cardiovascular Research Foundation Basel, that are unrelated to this work, and consultant fees from Janssen. Authors not named here have disclosed no competing interests.]

Answer: We have added the mentioned statement in the Competing Interests section in our revised cover letter.

Answer: Thank you for the opportunity to better explain the legal conditions of this COVID-19 study, which differ from "regular" studies. In order to allow, facilitate and accelerate urgently needed clinical studies during the early phase of the COVID-19 pandemic, the responsible ethics committee (Ethics Committee Nordwest- und Zentralschweiz (EKNZ), Hebelstrasse 53, 4056 Basel, Tel. 061 268 13 50, Fax 061 268 13 51, Email: eknz@bs.ch; EKNZ identifier 2020-00566) waived the need for a study-specific informed consent. Instead, they gave permission to include patients in this study based on the signature of an unspecific general consent, which allows to analyse clinical parameters and blood remains that are collected during clinical routine and to contact patients up to 30 days after hospital discharge. However, this general consent does not permit to make patient-level data publicly available, not even in an anonymised fashion, as these are highly sensitive and potentially identifying patient data from a single-centre study obtained during a short period of time. However, in case of a request for a scientific collaboration, data sharing could be allowed under the umbrella of a site-by-site data transfer agreement granting adequate data protection and confidentiality. We now state:

"Access to the data is restricted by the review board of the COVIVA Study. Data requests may be directed to Professor Raphael Twerenbold (raphael.twerenbold@usb.ch), or to Gian Völlmin (gian.voellmin@usb.ch) as non-author representative of the data access committee."

4. Thank you for submitting the above manuscript to PLOS ONE. During our internal evaluation of the manuscript, we found significant text overlap between your submission and the following previously published works, some of which you are an author.

Lampart M, Rüegg M, Jauslin AS, Simon NR, Zellweger N, Eken C, Tschudin-Sutter S, Bassetti S, Rentsch KM, Siegemund M, Bingisser R, Nickel CH, Osswald S, Kuster GM, Twerenbold R. Direct Comparison of Clinical Characteristics, Outcomes, and Risk Prediction in Patients with COVID-19 and Controls—A Prospective Cohort Study. Journal of Clinical Medicine. 2021; 10(12):2672. https://doi.org/10.3390/jcm10122672

Please revise the manuscript to rephrase the duplicated text, cite your sources, and provide details as to how the current manuscript advances on previous work. Please note that further consideration is dependent on the submission of a manuscript that addresses these concerns about the overlap in text with published work.

Answer: We have carefully rewritten the manuscript to minimize overlap with earlier publications. However, as already pointed out by the editor, some overlap in the methods section is inevitable and may persist. Please let us know if the present version does not fit your expectations.

 

Reviewer #1

This is an interesting study, one of the first trying to compare inflammatory makers in covid and other respiratory infections. However, some aspects of the study, especially the classification of other respiratory infections, could be better explained. 

INTRODUCTION

1. The immunological mechanisms underlying the increase / decrease of markers are explained in a simplistic way. I suggest expanding the introduction by adding more information (e.g., adding the normal values of the cited markers, explaining for each marker the (presumed) mechanism of increase/decreased both in covid / bacterial and viral infections.

For example, about leukocyte levels:

I suggest adding: “Leukocytosis is defined by an increase in the WBC count of more than 11,000 cells/microL.”

Line 92: “Leukocyte levels often increase during infections due to the activation of the immune system.” 

I suggest to better explain the concepts of neutrophilic leukocytosis in bacterial infection and lymphocytosis/lymphopenia in viral infection, so for example:

Leukocyte levels often increase during infections, due to the release of several molecules, as growth or survival factors, adhesion molecules and various cytokines released during activation of immune system. Most bacterial infections are associated with neutrophilic leukocytosis. Neutrophilia occurs from both upregulated bone marrow production and the release of neutrophils from the endothelium. Generally, most viruses lead to relative lymphocytosis, while only a few viruses causing lymphopenia, such as SARS-COV-2.

Answer: Thank you for your excellent comment. We have now integrated your valuable suggestions into the Introduction. Additionally, we expanded the Introduction to add more information about each inflammatory biomarker. 

2. Line 96-97: “as SARS CoV-2 binds to the angiotensin converting enzyme 2 (ACE2), which is located on most lymphocytes” This sentence is not enough specific to explain the mechanism of lymphopenia. 

I suggest to better specify the mechanisms of lymphopenia in covid-19 (for example, referring to Jafarzadeh A, et al. Lymphopenia an important immunological abnormality in patients with COVID-19: Possible mechanisms. Scand J Immunol. 2021)

Answer: Thank you for your excellent comment. We have now specified the mechanisms that could lead to lymphopenia and included the outstanding work you suggested.

“The causes for lymphopenia in COVID-19 have not yet been conclusively determined. Possible mechanisms include, but are not limited to, SARS-CoV-2-induced apoptosis of lymphocytes via the angiotensin converting enzyme 2, CRS-induced apoptosis of lymphocytes, and antibody-dependent killing of SARS-CoV-2-infected lymphocytes”

Page 6, Line 110-114

MATERIALS AND METHODS

3. Line 106: “included unselected patients aged 18 years and older 106 presenting with clinically suspected or confirmed SARS-CoV-2 infection”. Was the enrolment of patients consecutive? Please specify.

Answer: Indeed, the recruitment of the patients in this study was consecutive. Following the suggestion of this reviewer, we now describe this important characteristic more specifically in the study design section.

“The COronaVIrus surviVAl (COVIVA, ClinicalTrials.gov NCT04366765) is a prospective, observational cohort study including consecutive patients aged minimally 18 years presenting with clinically suspected or confirmed SARS-CoV-2 infection to the emergency department (ED) of the University Hospital Basel, Switzerland, during the first wave of COVID-19 pandemic between 23 March 2020 and 7 June 2020.”

Page 7, Line 124-128

4. Line 126: Why do you choose “leukocytes” and you don’t differentiate between neutrophils and lymphocytes? Please specify.

Answer: Thank you for this important comment. Unfortunately, lymphocyte counts, and particularly neutrophil counts are not available in our dataset, as they were not part of the routine laboratory panel in the ED of the recruiting centre. Therefore, we cannot further differentiate between leukocytes. We now mention the lack of white blood cell differential in the methods section.

“Blood samples were drawn in both cases and controls at time of ED presentation. CRP, ferritin, and leukocytes without further white blood cell differential were measured in fresh samples as part of clinical routine of the recruiting hospital”.

Page 8, Line 146-148

5. Line 132-133: Why treating physicians were not blinded for the other markers? Please specify.

Answer: Thank you very much for this important remark which gives us the opportunity to better describe the standard operating procedures in the recruiting ED. 

As state of the art to assess patients with dyspnoea and other symptoms suggestive of COVID-19, the treating physicians were dependent on established inflammatory laboratory parameters. These included in our setting the routine measurement of leukocytes as well as CRP. The blinding of these parameters would have potentially harmed the patients and negatively impacted patient management. In addition, due to its potential to predict the risk of acute respiratory distress syndrome, ferritin was added to the standard inflammation panel upon request of the intensive care physicians of the recruiting centre. In contrast, treating physicians were blinded for PCT and IL-6 as these inflammatory biomarkers were measured subsequently from blood samples stored in the dedicated biobank.

6. Line 158: Please add the reference to S1 (flowchart of the study). Is it possible to move the flowchart of the study from the supplementary material to the main text? the flow chart is well constructed and helps the reader to follow the text.

Answer: Thank you very much for this kind remark. We have now included the flow chart as Figure 1 in the main text. We adjusted the following figure numbers accordingly.

7. Line 158-161: How did the five trained physicians differentiate between viral and bacterial pneumonia? Did you use specific criteria (e.g., bacterium isolation on sputum sample, urinary antigen positivity for pneumococcus, virus isolation on multiplex PCR, lobar or interstitial pneumonia on chest x-ray)? Or the decision was only “clinical”? Please specify. This is a crucial point of the study design.

Is it possible to have some information relating to the identification of bacteria or other viruses?

Answer: Thank you very much for this important comment. We now describe in more details the distinction between bacterial pneumonia and viral respiratory infections, which was performed based on clinical interpretation.

“The distinction between bacterial pneumonia and viral respiratory infection was primarily based on clinical examination (e.g., rales, fever, tachypnoea) and particularly radiological findings (e.g., lobar or interstitial pneumonic infiltrates in the x-ray or CT scan of the lungs). No specific pathogen distinction to identify the underlying bacterium or virus (e.g., bacterium isolation on sputum sample, urinary antigen positivity for pneumococcus or legionella, virus isolation on multiplex PCR) was systematically performed as part of clinical routine and was therefore largely missing.”

Page 9, Line 182-188

In addition, we now comment on our approach in the limitation section: 

“Second, despite our efforts to differentiate viral from bacterial respiratory infections based on clinical examinations and radiological findings, we cannot guarantee that a small proportion of patients were misclassified due to the missing routine distinction of pathogens.”

Page 25, Line 407-410

RESULTS

8. Table 1: were coronary artery disease, prior myocardial infarction, atrial fibrillation and hypertension included in “cardiac disease”? please specify.

Answer: In our study, cardiac disease was defined as the composite of history of coronary artery disease, myocardial infarction, valvular cardiopathy, atrial fibrillation and congestive heart failure. Arterial hypertension was considered separately and did not account for cardiac disease. We now describe in more details in the legend of table 1 and table 2. Similarly, we also make a remark about pneumopathy which consists of COPD and Asthma.

9. In table 1 we can see that some patients have immunodeficiency (even if the cause of immunodeficiency is not specified). Do you think the immunodeficiency could altered the inflammatory response (and therefore consequently also the values of the analysed markers)? Please explain the choice to include immunodeficiency in the analyses.

Answer: We aimed to depict an all-comer analysis in unselected, consecutive patients presenting to the ED. Accordingly, we also included a low number of patients with immunodeficiency, which was mostly explained by the regular intake of oral corticosteroids at time of ED presentation. Unfortunately, the number of affected patients seems by far too low to explore subgroup analyses and draw robust findings.

10. Line 233-234. As you described first the hospitalized group, I suggest inverting the example because (e.g., CRP 2.3 mg/l (IQR 0.9-234.1) refers to non-hospitalized patients while 59.3 mg/l (IQR 31.5-126.9) refers to hospitalized patient. p<0.001. The same for line 237 and 238.

Answer: Thank you for your attentive reading of our work and rightfully commenting about the order of the given values. As suggested by this reviewer, we now have changed the respective sentences to increase clarity.

DISCUSSION

11. Discussion clearly reports the results without comparing them with other studies. Please add, if possible, more comparison with other studies such as those mentioned in the introduction.

Answer: As suggested by the reviewer, we now have compared our results with findings from other studies in more detail. 

12. Line 325-327: “This finding suggests that inflammatory reactions in COVID-19 are in general more pronounced than in other viral respiratory infections but less than in bacterial pneumonia”. This conclusion is speculative: the fact that the inflammatory markers are different in the three cases does not mean that the inflammatory mechanism in covid 19 is less pronounced. Please rephrase.

Answer: Thank you very much for your comment. As suggested, we have changed the wording of the respective paragraph and toned down our previous interpretation: 

“This finding corroborates the results from other studies reporting low leukocyte levels in COVID-19 (10,45,47,48). However, in contrast to most previous studies, our analyses allow to interpret such findings in direct comparison with other respiratory infections. Whether inflammatory reactions in COVID-19 are in general truly more pronounced than in other viral respiratory infections but less than in bacterial pneumonia remains speculative and needs to be addressed in future studies.”

Page 23, Line 355-361

13. Line 383: please add to the limit session also the risk of misclassification of bacterial and viral pneumoniae (the method of diagnosis is not clear).

Answer: We fully agree with this reviewer and now have added this important aspect to the limitation section.

“Second, despite our efforts to differentiate viral from bacterial respiratory infections based on clinical examinations and radiological findings, we cannot guarantee that a small proportion of patients were misclassified due to the missing routine distinction of pathogens.”

Page 25, Line 407-410

 

Reviewer #2

Thank you for the opportunity to review this article

The work of Lampart et al. is interesting and analysed the role of inflammatory factors in COVID-19 and other respiratory infections. The article outlines some clinically important aspects.

I have some observations:

1. In Materials and methods, line 106: “…infection to the ED of….” What does ED mean? Please put it in full

Answer: Thank you for noticing, we now define ED as emergency department in the Materials and Methods. 

2. In Materials and Methods how were other viral and bacterial infections (viral and bacterial controls) diagnosed? Please specify diagnostic methods and viral and bacterial diagnosis.

Answer: Thank you very much for this important comment. We now describe in more details the distinction between bacterial pneumonia and viral respiratory infections, which was performed based on clinical interpretation.

“The distinction between bacterial pneumonia and viral respiratory infection was primarily based on clinical examination (e.g., rales, fever, tachypnoea) and particularly radiological findings (e.g., lobar or interstitial pneumonic infiltrates in the x-ray or CT scan of the lungs). No specific pathogen distinction to identify the underlying bacterium or virus (e.g., bacterium isolation on sputum sample, urinary antigen positivity for pneumococcus or legionella, virus isolation on multiplex PCR) was systematically performed as part of clinical routine and was therefore largely missing.”

Page 9, Line 182-188

In addition, we now comment on our approach in the limitation section: 

“Second, despite our efforts to differentiate viral from bacterial respiratory infections based on clinical examinations and radiological findings, we cannot guarantee that a small proportion of patients were misclassified due to the missing routine distinction of pathogens.”

Page 25, Line 407-410

3. The bacterial control was significantly older and with more comorbidities than the others group. In the Author’s opinion, could this have contributed to the higher values of the inflammatory indices?

Answer: Thank you for this comment. We fully agree that patients with pneumonia are older and more comorbid than patients with COVID-19 and viral respiratory infections. The small size of our dataset unfortunately does not allow to perform extensive analyses including adjustment for potential confounders. Whether the higher age of patients with pneumonia may rather increase or decrease inflammatory response remains therefore unclear. We now list in the limitation section the missing adjustment for potential confounders such as age and comorbidities.

“Similarly, due to the rather small sample size, no inter-group adjustment for potential confounders (e.g., age, comorbidities) was applied.”

Page 26, Line 412-414

---

## [Decision Letter · Decision Letter 1]

13 May 2022

Clinical utility of inflammatory biomarkers in COVID-19 in direct comparison to other respiratory infections - A prospective cohort study

PONE-D-21-39440R1

Dear Dr. Lampart,

We’re pleased to inform you that your manuscript has been judged scientifically suitable for publication and will be formally accepted for publication once it meets all outstanding technical requirements.

Kind regards,

Paola Faverio

Academic Editor

PLOS ONE

Additional Editor Comments (optional):

The points raised by the Reviewers and myself have been sufficiently addressed.

Reviewers' comments:

Reviewer's Responses to Questions

**Comments to the Author**

1. If the authors have adequately addressed your comments raised in a previous round of review and you feel that this manuscript is now acceptable for publication, you may indicate that here to bypass the “Comments to the Author” section, enter your conflict of interest statement in the “Confidential to Editor” section, and submit your "Accept" recommendation.

Reviewer #2: All comments have been addressed

2. Is the manuscript technically sound, and do the data support the conclusions?

Reviewer #2: Yes

3. Has the statistical analysis been performed appropriately and rigorously? 

Reviewer #2: Yes

4. Have the authors made all data underlying the findings in their manuscript fully available?

Reviewer #2: Yes

5. Is the manuscript presented in an intelligible fashion and written in standard English?

Reviewer #2: Yes

6. Review Comments to the Author

Reviewer #2: (No Response)

7. PLOS authors have the option to publish the peer review history of their article (what does this mean?). If published, this will include your full peer review and any attached files.

Reviewer #2: No

---

## [Editor Report · Acceptance letter]

19 May 2022

PONE-D-21-39440R1 

Clinical utility of inflammatory biomarkers in COVID-19 in direct comparison to other respiratory infections - A prospective cohort study 

Dear Dr. Twerenbold:

I'm pleased to inform you that your manuscript has been deemed suitable for publication in PLOS ONE. Congratulations! Your manuscript is now with our production department. 

Kind regards, 

on behalf of

Dr. Paola Faverio 

Academic Editor

PLOS ONE